# Statistical estimation of spatial wave extremes for tropical cyclones from small data samples: validation of the STM-E approach using long-term synthetic cyclone data for the Caribbean Sea

Ryota Wada[1], Jeremy Rohmer[2], Yann Krien[3], and Philip Jonathan[4,5]

[1]The University of Tokyo, Tokyo, Japan
[2]BRGM, Orleans, France
[3]SHOM, DOPS/HOM/REC, Toulouse, France
[4]Shell Research Limited, London SE1 7NA, United Kingdom.
[5]Department of Mathematics and Statistics, Lancaster University LA1 4YF, United Kingdom.

**Correspondence:** Ryota Wada (`r_wada@k.u-tokyo.ac.jp`)

**Abstract.** Occurrences of tropical cyclones at a location are rare, and for many locations, only short periods of observations or hindcasts are available. Hence, estimation of return values (corresponding to a period considerably longer than that for which data is available) for cyclone-induced significant wave height (SWH) from small samples is challenging. The STM-E (space-time maximum and exposure) model was developed to provide reduced bias in estimates of return values compared to competitor approaches in such situations, and realistic estimates of return value uncertainty. STM-E exploits data from a spatial neighbourhood satisfying certain conditions, rather than data from a single location, for return value estimation.

This article provides critical assessment of the STM-E model for tropical cyclones in the Caribbean Sea near Guadeloupe for which a large database of synthetic cyclones is available, corresponding to more than 3,000 years of observation. Results indicate that STM-E yields values for the 500-year return value of SWH and its variability, estimated from 200 years of cyclone data, consistent with direct empirical estimates obtained by sampling 500 years of data from the full synthetic cyclone database; similar results were found for estimation of the 100-year return value from samples corresponding to approximately 50 years of data. In general, STM-E also provides reduced bias and more realistic uncertainty estimates for return values relative to single location analysis.

KEYWORDS: tropical cyclone spatial extremes synthetic storm return value

## 1 Introduction

Tropical cyclones (also named hurricanes or typhoons depending on the region of interest) are one of the deadliest and most devastating natural hazards that can significantly impact lives, economies and the environment in coastal areas. In 2005, hurricane Katrina, which hit New Orleans, was the most costly natural disaster of all time for the insurance sector, with losses totalling more than $10^{11}$ US dollars (Barbier 2015). In 2017, hurricanes Harvey, Irma and Maria caused record losses within just four weeks totalling more than $9 \times 10^{10}$ US dollars [1]. Tropical cyclones present multiple hazards, including large damaging

---

[1]https://www.munichre.com/en/risks/natural-disasters-losses-are-trending-upwards/hurricanes-typhoons-cyclones.html-1979426458

winds, high waves, storm surges, and heavy rainfall, as exemplified by Typhoon Hagibis in Japan (see context description in Dasgupta et al. 2020) or Cyclone Idai in Mozambique in 2019. [2]

Waves are one of the major hazards associated with tropical cyclones, of critical importance regarding marine flooding, especially for volcanic islands like those in the Lesser Antilles – North Atlantic ocean basin (Krien et al. 2015), in the Hawaiï – Northeast Pacific ocean basin (Kennedy et al. 2012), or in the Reunion Island – Southwest Indian ocean basin (Lecacheux et al. 2021). Here, the absence of a continental shelf and the steep coastal slopes limit the generation of high atmospheric storm surge, but increase the potential impact of incoming waves. Moreover, wind-waves propagate with little loss of energy over the deep ocean: this might potentially increase the spatial extent as well as time duration over which damaging coastal impacts occur during a tropical cyclone event (Merrifield et al. 2014); this contrasts with tropical cyclone-induced storm surge, which tends to be concentrated in the vicinity of the cyclone centre.

To help decision makers in diverse fields such as waste-water management, transport and infrastructure, health, coastal zone management, and insurance, one key ingredient is the availability of data for the frequencies and magnitudes of extreme cyclone-induced coastal significant wave heights SWH, e.g. estimates of 100-year return values (e.g. as illustrated for Reunion Island by Lecacheux et al. 2012: Figure 4). Yet, for many locations, only short periods of observations or hindcasts of tropical cyclones are available, which can be challenging for estimation of return values (corresponding to a period considerably longer than that for which data are available). For this purpose, a widely-used approach relies on the combination of synthetic cyclone track generation, wave modeling and extreme value analysis. The approach consists in the following steps: (1) tropical cyclones, extracted from either historical data (Knapp et al. 2010) or climate model simulations (Lin et al. 2012) are statistically resampled and modeled to generate synthetic, but realistic tropical cyclone records. Based on a Monte Carlo approach (Emanuel et al. 2006; Vickery et al. 2000; Bloemendaal et al. 2020), a tropical cyclone dataset with same statistical characteristics as the input dataset, but spanning hundreds to thousands of years, can then be generated; (2) for each synthetic cyclone, a hydrodynamic numerical model is used to compute the corresponding SWH over the whole domain of interest. An example of such a simulator is the Global Tide and Surge Model of Bloemendaal et al. (2019); (3) SWH values at the desired coastal locations are extracted. Extreme value analysis (Coles et al. 2001) can then be used to estimate the corresponding return values. As an illustration of the whole procedure, one can refer to the probabilistic hurricane-induced storm surge hazard assessment (including wave effects) performed by Krien et al. (2015) at Guadeloupe archipelago, Lesser Antilles.

Implementation of steps (1) and (2) can however be problematic. Generation of synthetic cyclones with realistic characteristics is a research topic in itself. Further, the hydrodynamic numerical model can be prohibitively costly to execute, limiting the number of model runs feasible, resulting in sparse, non-representative data for extreme value modelling. To overcome this computational burden, possible solutions can either be based on parametric analytical models (like the ones used by Stephens and Ramsay 2014 in the Southwest Pacific Ocean) or on statistical predictive models (sometimes called meta- or surrogate models, Nadal-Caraballo et al. 2020). However, such approaches can only be considered "approximations". The former parametric analytical models introduce simplifying assumptions regarding the physical processes involved. Statistical estimation is

---

[2]https://data.jrc.ec.europa.eu/dataset/4f8c752b-3440-4e61-a48d-4d1d9311abfa

problematic, since inferences must be made concerning extreme quantiles of the distribution of quantities such as SWH, using a limited set of data.

**Objective and layout**

In the present work, we aim to tackle the problem of realistic return value estimation for small samples of tropical cyclones using a recently-developed procedure named STM-E, which has already been successfully applied in regions exposed to tropical cyclones near Japan (Wada et al. 2018) and in Gulf of Mexico (Wada et al. 2020). STM-E exploits all cyclone data drawn from a specific geographical region of interest, provided that certain modelling conditions are not violated by the data. This means in principle that STM-E provides less uncertain estimates of return values than statistical analysis of cyclone data at a single location. To date however, the STM-E methodology has not been directly validated: the objective of the present work is therefore to provide direct validation of return values (in terms of bias and variance characteristics, for return periods $T$ of hundreds of years) from STM-E analysis using sample data for modelling corresponding to a much shorter period $T_0$ ($< T$) of observation, drawn from a full synthetic cyclone database corresponding to a very long period $T_L$ ($T_L > T$) of observation.

In the following sections, we present a motivating application in the region of the Caribbean archipelago of Guadeloupe, for which synthetic cyclone data are available for a period $T_L$ corresponding to more that 3,000 years. We use the STM-E method to estimate the $T = 500$-year return value for SWH, and its uncertainty, based on random samples of tropical cyclones corresponding to $T_0 = 200$ years of observation. This case will assess the performance of STM-E when reasonable sample sizes of extreme values are available for inference. In addition, we conduct the corresponding estimation for the $T = 100$-year return value for SWH, and its uncertainty, based on random samples of tropical cyclones corresponding to $T_0 = 50$ years of observation. This case is to assess the performance of STM-E under practical conditions, i.e. when the size of the sample of extreme data for analysis is relatively small. We compare estimates with empirical maxima from random samples corresponding to $T$ years of observation from the full synthetic cyclone data (covering $T_L$ years), and from standard extreme value estimates obtained using data (corresponding to $T_0$ years) from the specific location of interest only. Section 2 provides an outline of the motivating application. Section 3 describes the STM-E methodology. Section 4 presents the results of the application of STM-E to the region of the main island pair (Basse-Terre and Grande-Terre) of Guadeloupe. Discussion and conclusions are provided in Section 5.

## 2   Motivating application

The study area is located in a region of the Lesser Antilles (eastern Caribbean Sea) that is particularly exposed to cyclone risks (Jevrejeva et al. 2020) with several thousand fatalities reported since 1900 [3]. We focus on the French overseas region of Guadeloupe, which is an archipelago located in the southern part of the Leeward Islands (see Figure 1).

This French overseas region has been impacted by several devastating cyclones in the past, including the 1776 event (of category 5 according to the Saffir–Simpson scale, Simpson and Saffir 1974) which led to >6,000 fatalities (Zahibo et al. 2007),

---

[3]http://www.emdat.be

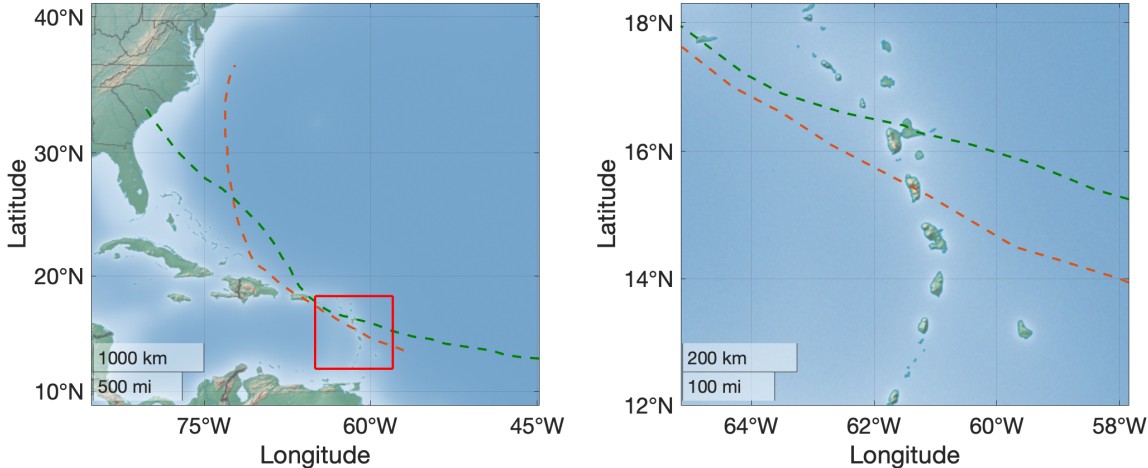

**Figure 1.** Regional setting. Left: Full domain. The red rectangle indicates the region where the diagnostic of the STM-E approach is performed. The orange and green tracks respectively represent those of Maria (2017) and Hugo (1989) cyclones (data extracted from Landsea and Franklin (2013) with cyclone status "Hurricane"). Right: Enlarged view of the diagnostic region. The red rectangle indicates the region in the vicinity of Guadeloupe archipelago where return values are estimated.

and the "Great Hurricane" of 1928 (Desarthe 2015) with >1,200 fatalities; the latter was probably the most destructive tropical cyclone of the 20th century. More recent destructive events include Hugo (in 1989, Koussoula-Bonneton 1994), and Maria (in 2017, which severely impacted Guadeloupe's banana plantations). The tracks of both Hugo and Maria are illustrated in Figure 1. Analysis of the HURDAT database (Landsea and Franklin 2013) reveals that approximately 0.6 cyclones per year passed within 400km of the study area on average for the period 1970-2019. Almost all events emanated from the south-east. More

than 80% of the events passed close to the northern and eastern coasts of Guadeloupe's main island pair.

To assess the cyclone-induced storm surge hazard, Krien et al. (2015) set up a modelling chain similar to that described in the introduction: they randomly generate cyclonic events using the approach of Emanuel et al. (2006), and compute SWH and total water levels for each event over a wide computational domain (45–65W, 9.5–18.3N) using the ADCIRC-SWAN wave-current coupled numerical model. . The interested reader can refer to Krien et al. (2015) for more implementation and validation detail.

The resulting SWH data are the basis of the current study to assess the performance of STM-E in estimating the $T$-year return value, from data corresponding to $T_0$ years of observation, for the cases $T_0 = 200, T = 500$ and $T_0 = 50, T = 100$.

In the present work, we use a total of 1971 synthetic cyclones passing nearby Guadeloupe (representative of 3,200 years, i.e. about 0.6 cyclone per annum) and the corresponding numerically calculated SWH. These results are used to derive empirically the 100-year SWH around the coast of Guadeloupe's main island pair for a smaller area of interest ($60.8 - 62.0°W, 15.8 -$

$16.6°N$; see Figure 2). These results are useful to assess flood risk at local scale, since they provide inputs of high resolution hydrodynamic simulations (see e.g. the use of wave over-topping simulations at La Reunion Island by Lecacheux et al. 2021). In the following, we analyse extreme SWH at 19 coastal locations around Guadeloupe's main island pair (on the 100m iso-

depth contour, see blue stars in Fig. 2), and at 12 locations along a line transect emanating to the north east from the island, corresponding to increasing water depth (see red triangles in Figure 2).

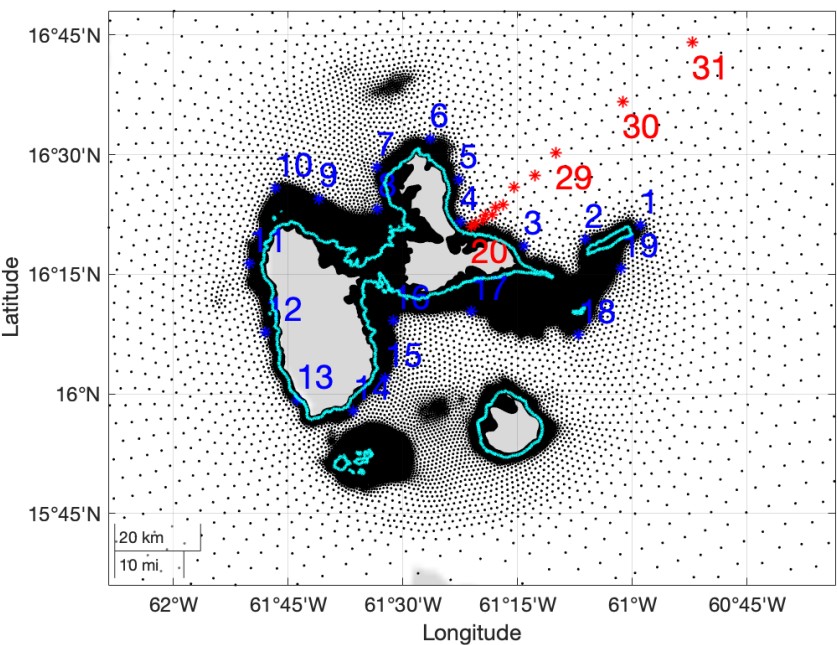

**Figure 2.** Illustration of Guadeloupe archipelago (administrative boundaries are outlined in light blue), showing calculation grid points (black dots), the selected numbered locations along the iso-depth contour at 100m (blue stars) and line transect (red stars). Calculation grid points are more dense in shallow waters. Water depths along the line transect with locations numbered 20-31 are 58m, 235m, 543m, 754m, 819m, 1206m, 1513m, 2350m, 3265m, 4059m, 5074m, 5586m.

To illustrate the SWH data, Figure 3 depicts the spatial distributions of maximum SWH per location for the four cyclones with the largest single values of SWH in the whole synthetic cyclone database. All cyclones propagate from the south-east to the north-west with intense storm severity near the cyclone track, which reduces quickly away from the track.

## 3    Methodology

In this section we describe the STM-E methodology used to estimate return values in the current work. Section 3.1 motivates
the STM-E approach, and Section 3.2 outlines the modelling procedure. Section 3.3 provides a discussion of some of the diagnostic tests performed to ensure that STM-E modelling assumptions are satisfied.

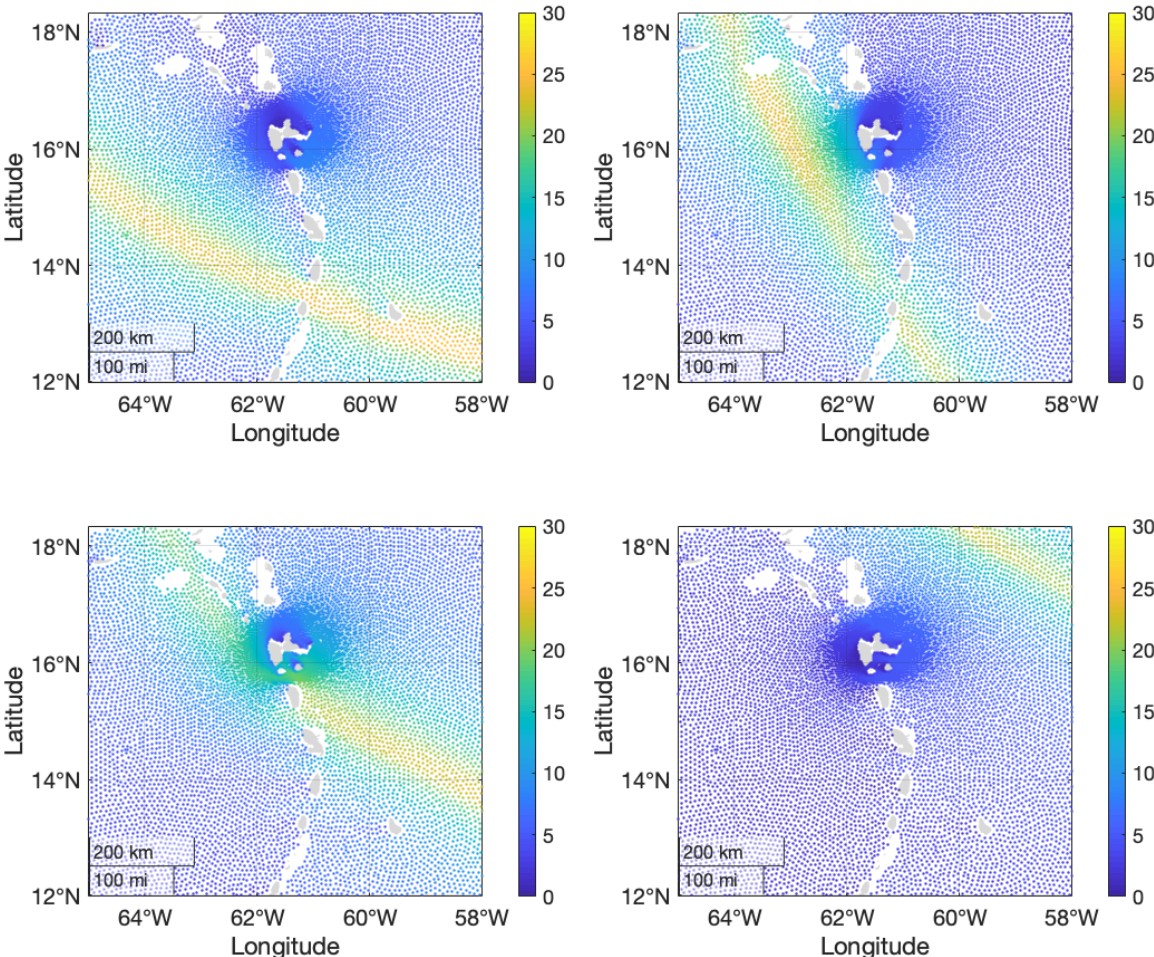

**Figure 3.** Spatial distributions of maximum SWH for the four largest synthetic cyclone events. Each panel gives the maximum SWH (over the period of the cyclone) per location.

### 3.1 Motivating the STM-E model

The STM-E procedure has been described in Wada et al. (2018) and Wada et al. (2020). The approach is intended to provide straightforward estimation of extreme environments over a spatial region, from a relatively small sample of rare events such as cyclones, the effects of any one of which do not typically influence the whole region. For each cyclone event, the space-time characteristics of the event are summarised using two quantities, the space-time maximum (STM) of the cyclone and the spatial exposure (E) of each location in the region to the event. For any cyclone, the STM is defined as the largest value of SWH

observed anywhere in the spatial region for the time period of the cyclone. The location exposure E is defined as the largest value of SWH observed at that location during the time period of the cyclone, expressed as a fraction of STM; thus values of E are in the interval $[0, 1]$.

The key modelling assumptions are then that (i) the future characteristics of STM and E over the region will be the same as those of STM and E during the period of observation, (ii) in future, at any location, it is valid to associate any simulated realisation of STM (under an extreme value model based on historical STM data) with any realisation of E (under a model for the distribution of E based on historical exposure data).

## 3.2 STM-E procedure

The steps of the modelling procedure are now described. The first three steps of the procedure involve isolation of data for analysis. (a) An appropriate region of ocean is selected. The characteristics of this region need to be such that the underpinning conditions of the STM-E approach are satisfied (as discussed further in Section 3.3). (b) For each tropical cyclone event occurring in the region, the largest value of SWH observed anywhere in the region for the period of the cyclone (STM) is retained. (c) Per location in the region, the largest value of SWH observed during the period of the cyclone, expressed as a fraction of STM, is retained as the location exposure E to the cyclone.

The next three steps of the analysis involve statistical modelling and simulation. (d) First, an extreme value model is estimated using the largest values from the sample of STM; typically, a generalised Pareto distribution (see e.g. Coles et al. 2001) is assumed. Then a model for the distribution of location exposure E is sought; typically we simply re-sample at random with replacement from the values of historical exposures for the location, although model-fitting is also possible. (e) Next, realisations of random occurrences of STM from (d), each combined with a randomly-sampled exposure E per location, permits estimation of the spatial distribution of SWH corresponding to return periods of arbitrary length. (f) Finally, diagnostic tools are used to confirm the consistency of simulations (e) under the model with historical cyclone characteristics.

## 3.3 Diagnostics for STM-E modelling assumptions

The success of the current approach relies critically on our ability to show that simplifying assumptions regarding the characteristics of STM and exposure are justified for the data to hand. In particular, the approach assumes that (i) the distribution STM does not depend on cyclone track, environmental covariates, space and time, and (ii) the distribution of exposure per location does not depend on STM, cyclone track, environmental covariates and time. Diagnostic tests are undertaken to examine the plausibility of these conditions for the region of ocean of interest for each application undertaken. Establishing the validity of the STM-E conditions is vital for credible estimation of return values. Section 5 of Wada et al. (2018) provides a detailed discussion of diagnostic tests that should be considered to judge that the STM-E conditions are not violated in any particular application. For example, the absence of a spatial trend in STM over the region can be assessed by quantifying the size of linear trends in STM along transects with arbitrary orientation in the region. This is then compared with a "null" distribution for linear trend, estimated using random permutations of the STM values. Illustrations of some of the diagnostic tests performed for the current analysis are given in Section 4 below.

Return value estimates from STM-E are also potentially sensitive to the choice of region for analysis. We assume that the extremal behaviour of STM can be considered homogeneous in the region, suggesting that the region should be sufficiently small that the same physics is active throughout it. However, the region also needs to contain sufficient evidence for cyclone events and their characteristics to allow reasonable estimation of tails of distributions for SWH per location. The absence of dependence between STM and E per location can be assessed by calculating the rank correlation between STM ($S$, a space-time maximum) and exposure ($E_j$, at location $j$, for locations $j = 1, 2, ..., p$) using Kendall's tau statistic. If the values of $S$ and $E_j$ increase together, the value of Kendall's tau statistic will be near to unity. If there is no particular relationship between $S$ and $E_j$, the value of Kendall's tau will be near zero. For large $n$, if $S$ and $E_j$ are independent, the value of Kendall's tau is approximately Gaussian-distributed with zero mean and known variance, providing a means of identifying unusual values which may indicate dependence between $S$ and $E_j$. An illustrative spatial plot of Kendall's tau is given in Section 4.

Finally, estimates from STM-E are potentially sensitive to the extreme value threshold $\psi_n$ (or equivalently the sample size $n$ of largest observations of STM) chosen to estimate the tail of the distribution of STM over the region. Results in Section 4 are reported for a number of choices of $n$ for this reason.

## 3.4   Modelling STM and estimating return values

Suppose we have isolated a set of $n_0$ values of STM using the procedure above. We use the largest $n \leq n_0$ values $\{s_i\}_{i=1}^n$, corresponding to exceedances of threshold $\psi_n$, to estimate a generalised Pareto model for STM, with probability density function

$$\Pr(S \leq s | S > \psi_n) = F_{S|\psi_S}(s) \tag{1}$$

$$\overset{\text{large } \psi_n}{\approx} F_{GP}(s) \quad = \quad 1 - \left(1 + \frac{\xi}{\sigma_n}(s - \psi_n)\right)^{-1/\xi} \quad \text{for } \xi \neq 0$$

$$= \quad 1 - \exp\left(-\frac{1}{\sigma_n}(s - \psi_n)\right) \quad \text{otherwise}$$

with shape parameter $\xi \in \mathbb{R}$ and scale parameter $\sigma_n > 0$. Choice of $n$ is important, to ensure reasonable model fit and bias-variance trade-off. The estimated value of $\xi$ should be approximately constant as a function of $n$ for sufficiently large $\psi_n$ and hence small $n$. The full distribution $F_S(s)$ of STM can then be estimated using

$$F_S(s) = \begin{cases} F_n^*(s) \text{ for } s \leq \psi_n \\ \tau_n + (1 - \tau_n) F_{S|\psi_n}(s) \text{ otherwise} \end{cases} \tag{2}$$

where $F_n^*(s)$ is an empirical "counting" estimate below threshold $\psi_n$, and $\tau_n$ is the non-exceedance probability corresponding the $\psi_n$, again estimated empirically.

Using this model, we can simulate future values $H_j$ of SWH at any location $j$, $(j = 1, 2, ..., p)$ in the region, relatively straightforwardly. Suppose that $E_j$ is the location exposure at location $j$, and $F_{E_j}$ its cumulative distribution function, estimated

empirically. Since then $H_j = E_j \times S$, the cumulative distribution function of $H_j$ can be estimated using

$$
\begin{aligned}
F_{H_j}(h) &= \mathbb{P}(H_j \leq h) \\
&= \int_s \mathbb{P}(E_j S \leq h \mid S = s) \, f_S(s) ds \\
&= \int_s \mathbb{P}(E_j \leq h/s) \, f_S(s) ds \\
&= \int_s F_{E_j}(h/s) f_S(s) ds
\end{aligned}
\tag{3}
$$

where $f_S(s)$ is the probability density function of STM, corresponding to cumulative distribution function $F_S(s)$ estimated in Equation 2.

## 4 Application of STM-E to cyclones SWH near Guadeloupe

The STM-E methodology outlined in Section 3 is applied to data for the neighbourhood of Guadeloupe's main island pair described in Section 2. The objective of the analysis is to estimate the $T$-year return value for SWH from $T_0$ years of data (for $(T_0, T)$ pairs (200,500) and (50,100)). First, details of the set-up of the STM-E analysis are provided in Section 4.1. Then, in Section 4.2, we describe two competitor methods included for comparison with STM-E. Section 4.3 then describes estimates for the 500-year return value on the 100m iso-depth contour around Guadeloupe's main island pair and the line transect introduced in Section 2, illustrated in Figure 2, using maximum likelihood estimation (see e.g. Hosking and Wallis 1987, Davison 2003). For comparison, Section 4.4 then provides estimates obtained using probability weighted moments (see e.g. Furrer and Naveau 2007, de Zea Bermudez and Kotz 2010a, de Zea Bermudez and Kotz 2010b). Section 4.5 describes some of the diagnostic tests undertaken to confirm that the fitted model is reasonable. Section 4.6 outlines inference for $T = 100$-year return value from data corresponding to $T_0 = 50$ years.

### 4.1 Details of STM-E application

The spatial region of interest is the neighbourhood of Guadeloupe's main island pair in the Caribbean Sea, corresponding to approximately longitudes $12° - 18°$N and latitudes $58° - 65°$W (see Figure 1). An initial analysis using Kendall's tau suggests the full region ($45° - 65°$W, $9.5° - 8.3°$N) shows dependency of STM and exposure, with stronger cyclones tending to pass through the western part of the region. However, if a very high threshold $\psi \approx 20$m were to be selected for analysis, reasonable decoupling of STM and E could be achieved, with relatively less intense tropical cyclones neglected. Since the focus of the current work is the ocean environment of Guadeloupe archipelago, a smaller region (see Figure 1, right panel) was defined. For this region, Kendall's tau indicated low dependence between STM and E for thresholds $\psi$ of 10m and above, as illustrated in the left panel of Figure 4.

The right panel of Figure 4 shows the location and magnitude of STM for each of the $n = 60$ largest cyclones observed in the region. There is no obvious spatial dependence between the size of STM and its location. In Section 3.3, we discuss the

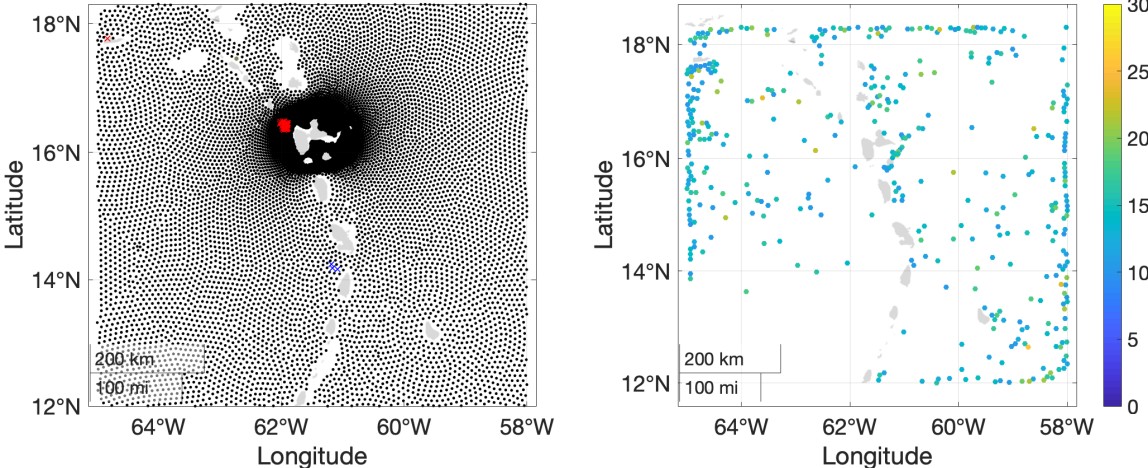

**Figure 4.** Diagnostics for STM-E. Left: plot of Kendall's tau for analysis region using a threshold of 10m. Each point corresponds to a location where SWH data is available and water depth exceeds 100m. The black dots indicate values of Kendall's tau within the 90% confidence interval. Red (blue) crosses indicate positive (negative) values of Kendall's tau exceeding the 90% confidence band. The percentage of recorded exceedances of the 90% confidence band for Kendall's tau is less than 10%. Right: Locations of all STMs exceeding 10m coloured by size of STM in metres.

use of rank correlation of STM along latitude-longitude transects as a means to quantify dependence in general. In fact, the Kendall's tau analysis illustrated in the left panel would also indicate any strong spatial dependence in STM; therefore, results

of the rank correlation analysis along latitude-longitude transects are not presented. We conclude that Figure 4 does not suggest that the modelling assumptions underlying STM-E are not satisfied.

The relatively large number of boundary STM values reflect occurrences of cyclones, the true STM locations of which occur outside the analysis region. For these events, the value of STM used for analysis is the largest value of SWH observed within the analysis region. In this sense, we are performing the STM-E analysis conditional on the choice of region. For example,

consider a cyclone for which the location of the STM value $s^*$ falls outside the region of interest. Then the conditional STM value $s$ for the cyclone (within the region) will of course be smaller than $s^*$; however, the cyclone's conditional exposure (assessed relative to the conditional STM $s$ for the region, rather than the full STM $s^*$) will consequently be larger.

Specific interest lies in the variation of extreme return value around Guadeloupe, and the rate of increase of return value with increasing water depth away from the coasts. It is known that SWH at a location is dependent on water depth, bathymetry and

coastlines, since e.g. ocean waves in shallow water are influenced by bottom effects, and since both wind and wave propagation can be weakened in the vicinity of coastlines. For this reason, two sets of locations were adopted for the detailed analysis reported here. The first set corresponds to 19 locations on an approximately iso-depth contour at 100m depth around the main island pair of Guadeloupe. This depth value is typically chosen to define the boundaries of the local scale high resolution flooding simulations. The second set corresponds to 12 locations on a line transect emerging approximately normally from

the north-east of the main island pair of Guadeloupe. We focus on the north-east neighbourhood, because it has the highest exposure to cyclones. The contour and transect are illustrated in Figure 2, and location numbers are listed.

Focus of the analysis is estimation of the $T = 500$-year return value for SWH on the iso-depth contour and line transect, based on $T_0 = 200$ years of data, to quantify the uncertainty in the 500-year return value using STM-E analysis. The following procedure was adopted. (a) Randomly select the appropriate number of cyclones (corresponding to $T_0$ years of observation) 230 from the $T_L$ years of synthetic cyclones. (b) Identify the largest $n$ values of STM in the sample, for $n = 20, 30, 40, 50$ and 60 (corresponding to lowering the extreme value threshold). (c) Estimate a model for the distribution of STM using maximum likelihood estimation or the method of probability weighted moments. (d) Estimate the empirical distribution of exposure E per location on the iso-depth contour and line transect. (e) Estimate the 500-year return value as the quantile of the distribution $F_{H_j}$ of significant wave height at location $j$ with non-exceedance probability $1 - (T_0/n)/T$. Finally, the whole procedure (a)-(e) is 235 repeated 100 times to quantify the uncertainty in the $T$-year return value.

Figure 5 illustrates the tails of the distribution of STM from the largest 30 values of STM from each of 100 random samples corresponding to 200 years, and from the full sample of synthetic cyclones. It can be seen that the 500-year return value for STM lies in the region (20,30)m. Typical distributions of exposure E per location are given in Figure 11, and discussed in due course.

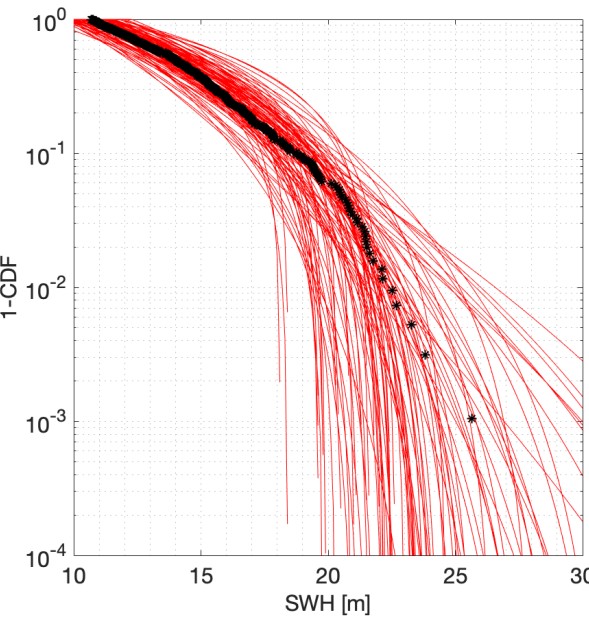

**Figure 5.** Variability of tail of distributions for STM on log scale. Each of the 100 red lines is estimated from a sample size 30 of largest STM values from a random sample of 124 cyclones corresponding to 200 years of observation. The black points indicate the corresponding empirical distribution of STM from the full synthetic cyclone data.

## 4.2 Benchmarking against the full cyclone database and single-location analysis

One obvious feature of the current synthetic cyclone database is that it corresponds to long time period relative of $T_F = 3,200$ years, much longer than the return period of $T = 500$ years being estimated in the current analysis. Thus, we are able to estimate the 500-year return value at any location using the full synthetic cyclone data, by simply interpolating the 6th and 7th largest values, corresponding to the non-exceedance probability in 500 years. This provides a direct empirical estimate.

From previous work, a key advantage found using the STM-E approach is that it provides less uncertain estimates at a location compared with conventional "single location" analysis. We wish to demonstrate in the current work that this is also the case. For this reason, we also calculate estimates for comparison with those from STM-E, based on independent analysis of cyclone data from each location of interest. The procedure is as follows. (a) Randomly select the appropriate number of cyclones (corresponding to $T_0$ years) from the $T_F$ years of synthetic cyclones for a single location (indexed by $j$). (b) Identify the largest value of SWH per cyclone, and call this the peak SWH. (c) Identify the $n$ values of peak cyclone SWH in the sample, for $n = 20, 30, 40, 50$ and $60$. (d) Estimate a model for the distribution of peak SWH using maximum likelihood estimation or the method of probability weighted moments. (e) Estimate the $T$-year return value as the quantile of the distribution of peak significant wave at location $j$ with non-exceedance probability $1 - (T_0/n)/T$. Finally, repeat (a)-(e) 100 times to quantify the uncertainty in the $T$-year return value.

## 4.3 Maximum likelihood estimation

Figure 6 illustrates the 500-year return value for SWH using maximum likelihood estimation for locations on the 100m iso-depth contour around Guadeloupe's main island pair, with location numbers given in Figure 2. The figure caption gives relevant details of the figure layout. Across the 19 locations considered, the 500-year return value is estimated using STM-E (blue), single-location (red) and full synthetic cyclone data (black); in general, there is good agreement between estimates per location. Bias characteristics for single-location and STM-E estimates are relatively similar in general. It can be seen however (from the longer red whiskers) that the uncertainty in single-location estimates is greater in general from the corresponding STM-E estimates.

As the number points used for STM-E estimation per location increases, there is evidence for reduction in the uncertainty with which the return value is estimated, as might be expected. However, there is also some evidence for a small increase in the mean estimated return value. This is explored further in Section 4.5. There is very little corresponding evidence for reduced uncertainty in the single-location analysis. There are more outlying estimates of return value for single-location analysis than for STM-E.

The corresponding results for the line transect analysis using maximum likelihood estimation is given in Figure 7. The general characteristics of this figure are similar to those of Figure 6. The return value increases as would be expected with increasing water depth. Single-point estimates are more variable that those from STM-E. Biases appear to be relatively small and similar for STM-E and single-location estimates. There is little evidence that the STM-E median estimate increases with

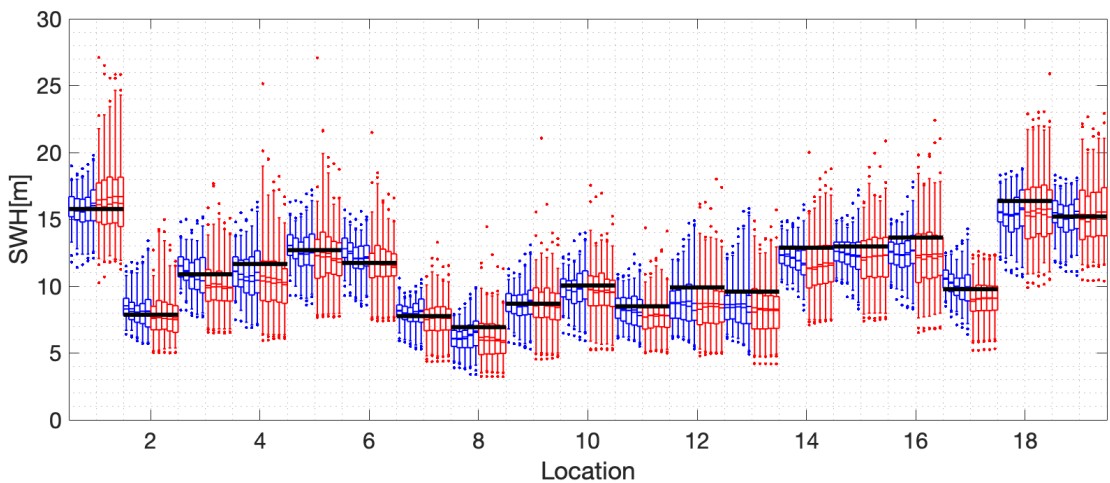

**Figure 6.** 500-year return value for SWH using maximum likelihood estimation for the 100m iso-depth contour. The x-axis gives the reference numbers of the 19 locations on the contour. Location numbers are given in Figure 2. Corresponding to each location, the blue box-whiskers summarise the estimated return value from STM-E for different sample sizes 20, 30, 40 50 and 60; the red box-whiskers summarise the estimated return values from single-location analysis for different sample sizes. For each blue-red cluster of box-whiskers corresponding to a specific location, return value estimates for the increasing sequence of sample sizes are plotted sequentially outwards from the centre of the cluster. For all box-whiskers, the box represents the inter-quartile interval, the median and mean are shown are solid and dashed lines. Whiskers represent the 2.5% to 97.5% interval. Exceedances of this 95% interval are shown as dots. The black horizontal line for each location corresponds to the empirical estimate of the return value obtained directly from the full synthetic cyclone data for that location.

increasing sample size. We infer from the analysis that water depth has little effect on the performance of the STM-E approach.

## 4.4 Results estimated using probability weighted moments

Estimates for the 500-year return value on the iso-depth contour, obtained using the method of probability weighted moments, are shown in Figure 8. The behaviour of STM-E and single-location estimates shown is very similar to that illustrate for maximum likelihood estimation in Figure 6.

Results for the line transect using probability weighted moments are given in Figure 9; again, the figure shows similar trends to Figure 7. There is some evidence that the STM-E median estimate increases with increasing sample size, and that this reduces bias.

## 4.5 Assessment of model performance

Comparing box-whisker plots from centre to left for each location in the figures in Sections 4.3 and 4.4 suggests that there is sometimes a small increasing trend in return value estimates from STM-E as a function of increasing sample size for inference.

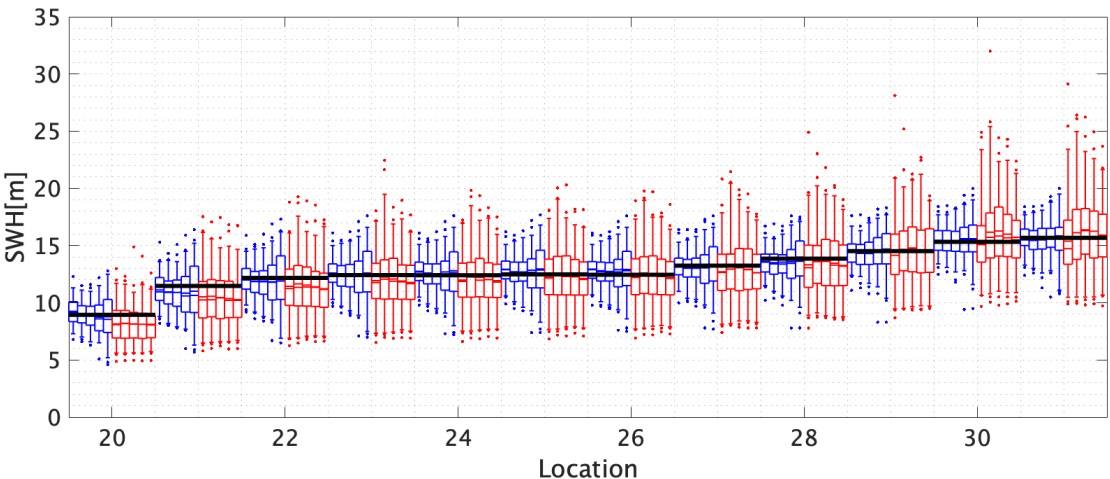

**Figure 7.** 500-year return value for SWH using maximum likelihood estimation for the line transect. The x-axis gives the reference numbers of the 12 locations on the transect. Briefly, for each location, blue and red box-whiskers summarise the estimated return value from STM-E and single-location analysis respectively; see caption of Figure 6 for other details. The black horizontal lines for each location correspond to the empirical estimate of the return value obtained directly from the full synthetic cyclone data for that location. Water depths at the 12 line transect locations are given in the caption to Figure 2.

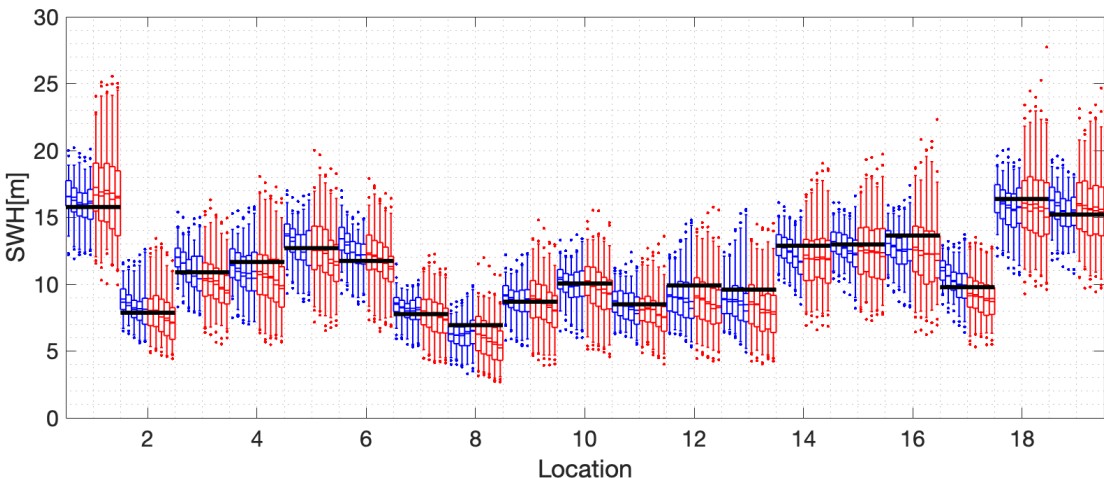

**Figure 8.** 500-year return value for SWH estimated using probability weighted moments for the 100m depth contour. The x-axis gives the reference numbers of the 19 locations on the contour. Briefly, for each location, blue and red box-whiskers summarise the estimated return value from STM-E and single-location analysis respectively; see caption of Figure 6 for other details. The black horizontal lines for each location corresponds to the empirical estimate of the return value obtained directly from the full synthetic cyclone data for each location.

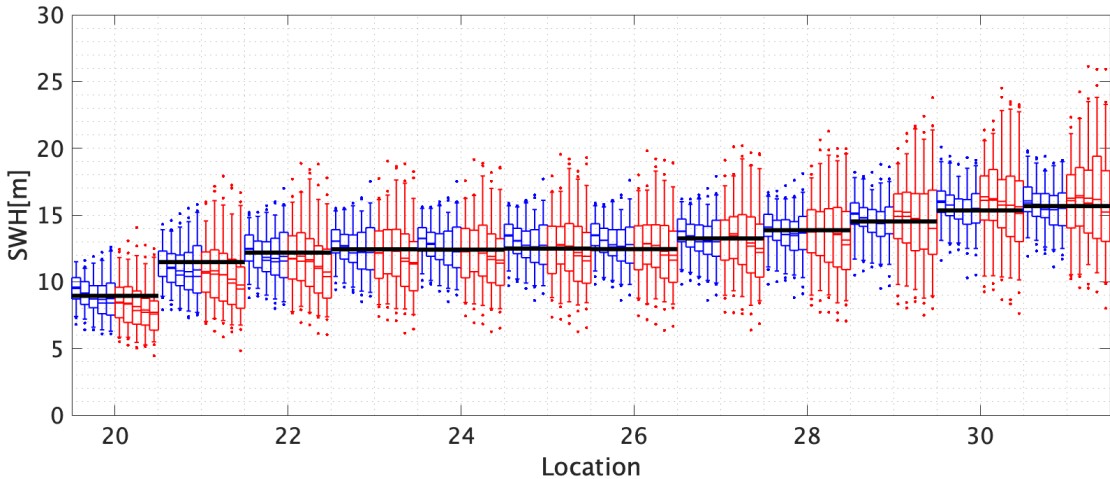

**Figure 9.** 500-year return value for SWH estimated using probability weighted moments for the line transect. The x-axis gives the reference numbers of the 12 locations on the transect. Briefly, for each location, blue and red box-whiskers summarise the estimated return value from STM-E and single-location analysis respectively; see caption of Figure 6 for other details. The black horizontal lines for each location corresponds to the empirical estimate of the return value obtained directly from the full synthetic cyclone data for each location. Water depths at the 12 line transect locations are given in the caption to Figure 2.

We investigate the trend further here. Figure 10 gives estimates for the 500-year return value of space-time maximum STM
(as opposed to the full STM-E estimate for SWH) as a function of sample size used for estimation, using maximum likelihood
estimation (blue) and probability weighted moments (red). Also shown in black is the empirical estimate of the 500-year STM
return value obtained directly from the synthetic cyclone data. The figure shows a number of interesting effects. Firstly, STM
estimates from both maximum likelihood and the probability weighted moments increase with increasing sample size $n$, and
that this effect is more pronounced for probability weighted moments. As a result, the bias of estimates using probability
weighted moments is considerably larger than that from maximum likelihood estimation for sample size of 60. The uncertainty
of estimates from probability weighted moments is also somewhat larger than that from STM-E.

A number of studies in the literature compare the performance of different methods of estimation of extreme value models.
The method of probability weighted moments is known to perform relatively well relative to maximum likelihood estimation
for small samples (see, e.g., Jonathan et al. (2021), Section 7 for a discussion). For small samples, for example, maximum
likelihood estimation is known to underestimate the generalised Pareto shape parameter, and over-estimate the corresponding
scale parameter, leading to bias in return value estimates. The results in Figure 10 indicate that, if anything, maximum likelihood
estimation performs somewhat better than the method of probability weighted moments for the current application. Regardless,
the trends in Figure 10 serve to illustrate the importance of performing the STM extreme value analysis with great care,
particularly for small sample sizes.

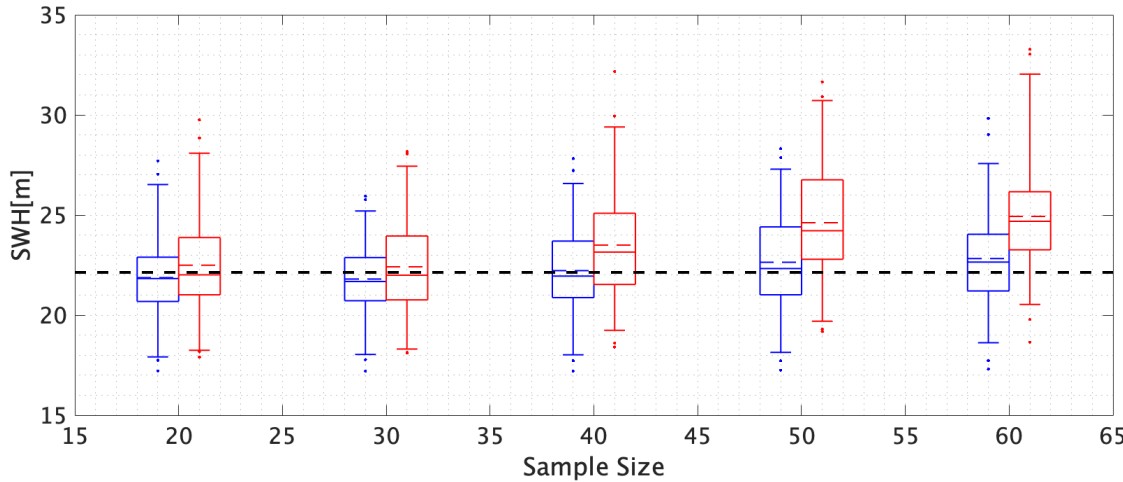

**Figure 10.** The effect of sample size on estimates of return values for STM. Box-whiskers summarise estimates for the 500-year STM return value based on maximum likelihood estimation (blue) and the method of probability weighted moments (red), for different sample sizes. The black dashed line gives an empirical estimate of the return value obtained directly from the synthetic cyclone data.

One of the assumptions underpinning the STM-E approach is that the exposure distribution at a location is not dependent on the magnitude of STM. We investigate this further here. Our aim is to show that the empirical cumulative distribution function for exposure (henceforth ECDF for brevity) corresponding to the largest and smallest STMs are typical of ECDFs in general, and are in no way special relative to EDCFs corresponding to other cyclones. We can quantify the difference between two ECDFs using the Kullback–Leibler divergence (KL). We proceed to estimate the "null" distribution of KL using 1,000 sets of

randomly-selected pairs of ECDFs. In addition, we calculate the Kullback-Leibler divergence (Liese and Vajda (2006)) $KL^*$ for the pair of ECDFs corresponding to cyclones with the largest and smallest STMs. If there is no dependence of ECDF on STM, then the value of $KL^*$ should correspond to a random draw from the null distribution of KL. The left-hand panel of Figure 11 illustrates the null distribution of KL at Location 21, for sample size 20, together with the corresponding value of $KL^*$. We note that the value of $KL^*$ is not extreme in the null distribution. In the right-hand panel of Figure 11, the empirical

cumulative distribution function of the non-exceedance probability of $KL^*$ (in the corresponding null distribution) is estimated over all locations and sample sizes. The approximate uniform density found, suggests indeed that $KL^*$ corresponds to a random draw from the null distribution; a Kolmogorov-Smirnoff test on the data suggested that it was not significantly different to a random sample from a uniform distribution on $[0, 1]$

       Complementary analyses (not shown) evaluated $KL^*$ for ECDFs corresponding to the largest two STMs in the data, and

(separately) for ECDFs corresponding to the smallest two STMs. Results again indicated that both of these choices for $KL^*$ could be viewed as random in their null distributions. Since exposure distribution at a location is not dependent on the magnitude of STM, we assume the overall performance of STM-E is mainly governed by the estimation of STM. In shallow water,

where waves are subject to breaking, it would be expected that the exposure distribution would be dependent on STM and therefore the validity of the method should be more carefully checked using the approaches described in Sect. 3.3.

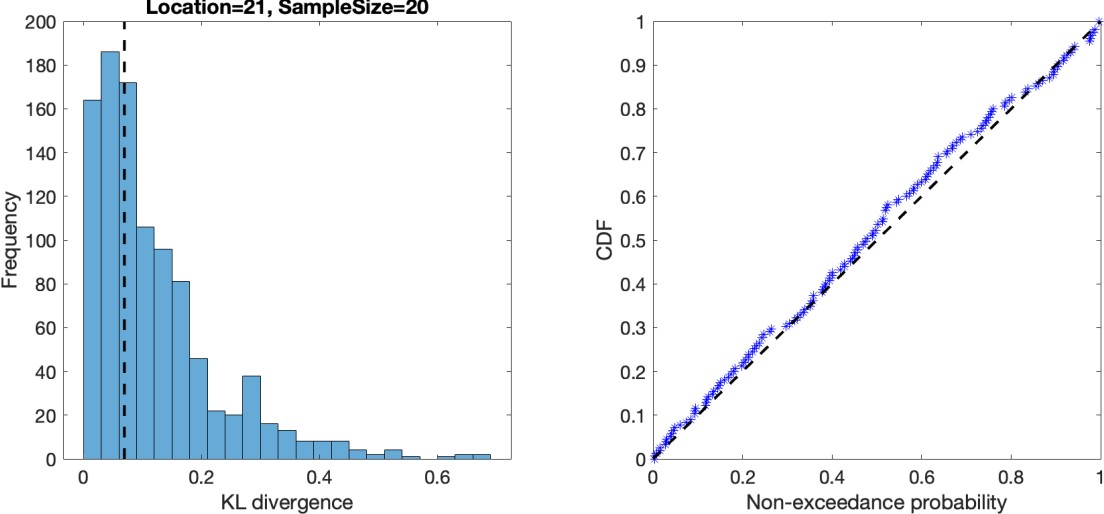

**Figure 11.** Left: Histogram of KL from random pairs of empirical distribution functions for exposure (corresponding to the "null" distribution of KL), together with $KL^*$ for Location 21 with sample size 20. Right: QQ-plot of the non-exceedance probability of $KL^*$ (in the corresponding null distribution for KL) over all locations and sample sizes.

## 4.6 Model performance for smaller sample sizes

Here we repeat the analysis in Sections 4.1-4.5 above for the $T = 100$-year return value for SWH on the iso-depth contour and line transect, based on $T_0 = 50$ years of data. The typical number of tropical cyclone events occurring in 50 years is approximately 30, already corresponding to a very small sample size for extreme value analysis. We retain the largest $n$ values of STM in the sample, for $n = 10, 15$ and 20 for this analysis. The overall performance of STM-E estimates, relative to those from single-location analysis and an empirical estimate from the full synthetic cyclone data is summarised in Figure 12 for the line transect, using the method of probability weighted moments (and see also Table 2 in the next section for a summary including estimates using maximum likelihood). The figure's features are similar to those of figures discussed earlier. Estimates from STM-E show lower bias and reduced uncertainty relative to those from single location analysis. There is slight underestimation of the return value, but the empirical estimate sits comfortably within the 25%-75% uncertainty band (corresponding to the "box" interval). The corresponding plots (not shown) for the iso-depth contour, and for estimation using maximum likelihood, are similar.

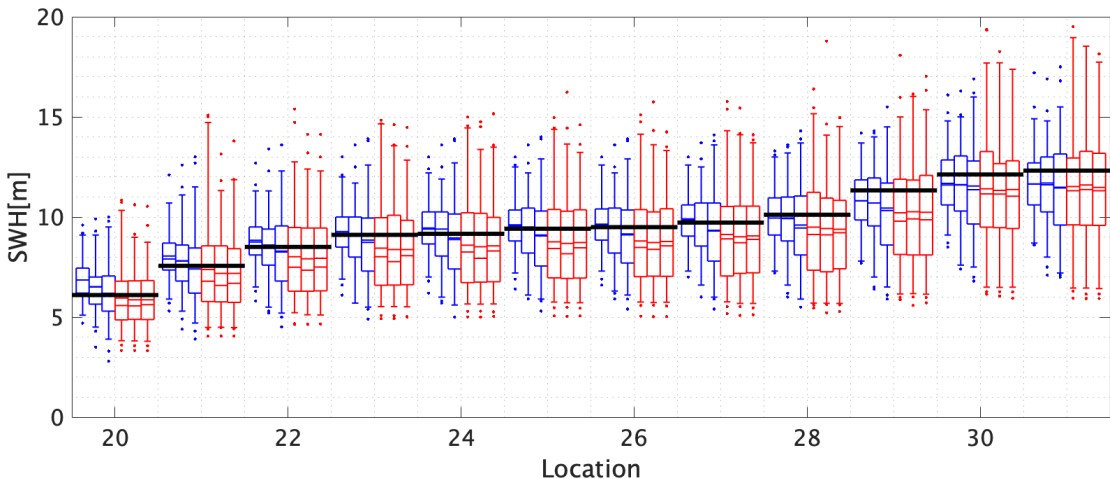

**Figure 12.** 100-year return value for SWH using probability weighted moments for the line transect. The x-axis gives the reference numbers of the 12 locations on the transect. Briefly, for each location, blue and red box-whiskers summarise the estimated return value from STM-E and single-location analysis respectively; see caption of Figure 6 for other details. The black horizontal lines corresponds to the empirical estimate of the return value obtained directly from the full synthetic cyclone data for each location. Water depths at the 12 line transect locations are given in the caption to Figure 2.

## 5    Discussion

This work considers the estimation of $T$-year return values for SWH over a geographic region, from small sets of $T_0$ years of synthetic tropical cyclone data, using the STM-E (space-time maxima and exposure) methodology. We assess the methodology by comparing estimates of the $T$-year return value ($T > T_0$) for locations in the region from STM-E, with those estimated directly from a large database corresponding to $T_L$ ($> T$) years of synthetic cyclones. We find that STM-E provides estimates of the $T = 500$-year return value from $T_0 = 200$ years of data in the region of Guadeloupe archipelago with low bias. We also compare STM-E estimates of $T$-year return values for locations in the region with those obtained by extreme value analysis of data (for $T_0$ years) at individual locations. We find that the uncertainty of STM-E estimates is lower than that of single-location estimates. Comparison of the performance of inferences for the $T = 100$-year return value from $T_0 = 50$ also suggests STM-E outperforms single-location analysis.

For reasonable application of the STM-E approach, it is important that characteristics of tropical cyclones over the region under consideration satisfy a number of conditions. These conditions are shown not to be violated for a region around Guadeloupe archipelago, but that use of the STM-E method over a larger spatial domain would not be valid (see e.g. Wada et al. 2019). This demonstrates that selection of an appropriate geographical region for STM-E analysis is critical to its success. Once such a region is specified, we find that STM-E provides a simple but principled approach to return value estimation within the region from small samples of tropical cyclone data.

Return value estimates from STM (see e.g. Figure 10) show a small increasing bias with increasing sample size for extreme value estimation. However, the resulting bias in full STM-E return values is small. Corresponding estimates based on single-location analysis also show relatively small but increasing negative bias with increasing sample size. In the present work, the tail of the distribution of STM was estimated by fitting a generalised Pareto model, using either maximum likelihood estimation or the method of probability weighted moments. Estimates for extreme quantiles of STM using either approach are in good agreement.

Table 1 summarises the performance of STM-E and single-location analysis in estimation of the bias and uncertainty of the 500-year return value, relative to empirical estimates from the full synthetic cyclone data, for analysis sample sizes of $n$=20, 30, 40, 50 and 60. Bias $B(n;\texttt{Mth})$ and uncertainty $U(n;\texttt{Mth})$ are estimated as average characteristics over all $|\mathcal{L}| = 31$ locations $\ell \in \mathcal{L} = \{1, 2, ..., 31\}$ on the iso-depth contour and line transect corresponding to the relevant sample size, using the expressions

$$B(n;\texttt{Mth}) = \frac{1}{|\mathcal{L}|} \sum_{\ell \in \mathcal{L}} \left( \tilde{h}(\ell;n,\texttt{Mth}) - \tilde{h}_0(\ell;n) \right), \qquad U(n;\texttt{Mth}) = \frac{1}{|\mathcal{L}|} \sum_{\ell \in \mathcal{L}} \left( \frac{r(\ell;n,\texttt{Mth})}{r_0(\ell;n)} - 1 \right). \qquad (4)$$

Here, $\tilde{h}(\ell;n,\texttt{Mth})$ and $\tilde{h}_0(n)$ correspond to the mean 500-year return value estimated using sample size $n$ from inference method $\texttt{Mth}$ (either maximum likelihood or probability-weighted moments) and directly from the full synthetic cyclone database; $r(\ell;n,\texttt{Mth})$ and $r_0(n)$ are the corresponding 50% uncertainty bands. The table summarises the findings presented pictorially in Figures 6-8. In terms of bias, STM-E and single-location estimates underestimate the return value on average. STM-E is less biased than single location estimates except for sample sizes 20 and 30 using probability weighted moments. STM-E also provides estimates of the 500-year return value with higher precision than the single-location analysis.

Table 2 provides the corresponding summary for estimation of the $T = 100$-year return value from a sample corresponding to approximately $T_0 = 50$ years of data. Features are similar to those of Table 1.

An appropriate choice of sample size $n$ for STM-E analysis is likely to be related to the size $n_0$ of the full sample available, and the period $T_0$ to which the sample corresponds. For example, in the current work for $T = 500$ years, $n = 20$ and $n_0 = 124$ is approximately equivalent to the largest 15% of cyclones for the sample period $T_0 = 200$ year. That is, the smallest cyclone considered in the $n = 20$ STM-E model has a return period of the order of 10 years. With $n = 60$, we use approximately half the sample for STM-E analysis, and the smallest cyclone in the STM-E analysis has a return period of the order of 3 years. In the case $T = 100$ years, $T_0 = 50$ and $n_0 \approx 30$, we found that STM-E performance was still reasonable using $n = 12, 15$ and 20.

Inferences from the current work confirm the findings of previous studies (Wada et al. 2018, Wada et al. 2020) that STM-E provides improved estimates of return values compared to statistical analysis at a single location. From an operational perspective, STM-E is useful for regions like the south-west Pacific ocean (McInnes et al. 2014) or Indian Ocean basin (Lecacheux et al. 2012) where cyclone-induced storm wave data is limited. For such locations, STM-E achieves low bias and higher precision, and should be preferred to the single-location approach.

]

**Table 1.** Performance of STM-E and single-location analysis in estimation of the bias and uncertainty of the 500-year return value, relative to empirical estimates from the full synthetic cyclone data, for analysis sample sizes of 20, 30, 40, 50 and 60 all extracted from 200-year data set. Bias $B$ is assessed as the average difference (over the iso-depth contour and line transect analyses) between the mean STM-E (or single-location) estimate and the return value estimate from the full cyclone database. Similarly, uncertainty $U$ is assessed in terms of the average width of the 50% uncertainty band of the STM-E (or single-location) estimate.

| Maximum likelihood | $n = 20$ | 30 | 40 | 50 | 60 |
|---|---|---|---|---|---|
| Bias mean STM-E | −0.073 | −0.171 | −0.225 | −0.198 | −0.018 |
| Bias median for single location | −0.437 | −0.225 | −0.246 | −0.333 | −0.419 |
| 50% intervals STM-E | 2.589 | 2.200 | 1.734 | 1.589 | 1.529 |
| 50% intervals for single location | 2.842 | 2.971 | 3.025 | 2.915 | 2.778 |
| **Probability weighted moments** | $n = 20$ | 30 | 40 | 50 | 60 |
| Bias mean STM-E | −0.199 | −0.263 | −0.158 | −0.109 | −0.381 |
| Bias median for single location | −0.158 | −0.096 | −0.246 | −0.494 | −0.713 |
| 50% intervals STM-E | 2.487 | 2.053 | 1.773 | 1.650 | 1.594 |
| 50% intervals for single location | 3.054 | 3.154 | 3.261 | 3.377 | 3.436 |

*Code availability.* MATLAB code for the analysis is provided on GitHub at Wada et al. (2021).

*Data availability.* Sample wave data for the analysis are available on Zenodo at Krien et al. (2021).

*Author contributions.* Conceptualization (JR, RW, PhJ), Cyclone and wave modelling (YK, JR), Statistical methodology and modelling (RW, PhJ), Writing (PhJ, JR, RW, YK)

*Competing interests.* The authors declare that they have no conflict of interest.

*Acknowledgements.* Numerical simulations were conducted using the computational resources of the C3I (Centre Commun de Calcul Intensif) in Guadeloupe. JR acknowledges the financial funding of the Carib-Coast INTERREG project (https://www.interreg-caraibes.fr/carib-coast). MATLAB code for the analysis is provided on GitHub at Wada et al. (2021). Sample wave data for the analysis are available on Zenodo at Krien et al. (2021).

**Table 2.** Performance of STM-E and single-location analysis in estimation of the bias and uncertainty of the 100-year return value, relative to empirical estimates from the full synthetic cyclone data, for analysis samples of size of 10, 15, and 20 all extracted from 50-year data set. Bias $B$ is assessed as the average difference (over the iso-depth contour and line transect analyses) between the mean STM-E (or single-location) estimate and return value estimated from the full cyclone data. Similarly, uncertainty $U$ is assessed in terms of the average width of the 50% uncertainty band of the STM-E (or single-location) estimate.

| **Maximum likelihood** | $n = 10$ | 15 | 20 |
|---|---|---|---|
| Bias mean STM-E | $-0.476$ | $-0.234$ | $-0.087$ |
| Bias mean for single location | $-0.744$ | $-0.787$ | $-0.771$ |
| 50% intervals STM-E | 2.545 | 2.089 | 1.705 |
| 50% intervals for single location | 3.071 | 3.044 | 3.005 |

| **Probability weighted moments** | $n = 10$ | 15 | 20 |
|---|---|---|---|
| Bias mean STM-E | $-0.740$ | $-0.246$ | $-0.017$ |
| Bias mean for single location | $-0.770$ | $-0.737$ | $-0.798$ |
| 50% intervals STM-E | 2.603 | 2.105 | 1.857 |
| 50% intervals for single location | 3.087 | 3.029 | 3.138 |

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
