# Peer review of "Statistical estimation of spatial wave extremes for tropical cyclones from small data samples: validation of the STM-E approach using long-term synthetic cyclone data for the Caribbean Sea"

_Natural Hazards and Earth System Sciences, 2021_

## Author Comment (AC1)

**Rebuttal for Reviewer #2**

**Dear Reviewer #2,**

**Overall review**
The paper presents a novel approach to predicting extreme wave conditions, the methodology appears to be rigorous and the study of high quality. The subject will be of interest to readers and is recommended for publication following consideration of the comments below.

*>> Thank you for the encouraging comments. We have considered your suggestions for revision individually. Please find our correspondence below (in italic).* Red text *indicates the actual modification on the manuscript. We hope you find them acceptable.*

**General comments:**
The main test is to compare 500 year return values predicted from 200 years of data. Was there a particular reason why the 200 years of data was selected? I would have thought that a smaller length of data, say 50 years typically the length of record available for cyclone, would provide a sterner test and may reveal larger differences between STM-E and single location analysis. That said, the paper is valid and worthwhile as it stands.

*>> Thank you for the comment. We agree with your comment and decided to add an additional analysis of 100-year return level estimation. Our initial aim was to validate methodology with long data, i.e. estimation with large enough samples. The case for 500-year return level estimation suggested that extracting the largest samples from around ⅓ of the tropical cyclones during the observation period is necessary to eliminate non-extreme samples.*
*Based on this suggestion, we provide proof of STM-E in a more realistic / practical situation, i.e. estimation of 100-year return level from 50 years data period choosing 10~ 20 top cyclones. The results for this analysis was generally the same as what we found in the 500yr return level case.*
*The results and discussion made from the analysis is provided in the manuscript as follows.*

[Modified Abstract: P1 Line 11]
similar results were found for estimation of the 100-year return value from samples corresponding to approximately 50 years of data.

[Modified Introduction (Objective and Layout): P3 Line 69]
This case will assess the performance of STM-E when reasonable sample sizes of extreme values are available for inference. In addition, we conduct the corresponding estimation for the T = 100- year return value for SWH, and its uncertainty, based on random samples of tropical cyclones corresponding to T0 = 50 years of observation. This case is to assess the performance of STM-E under practical conditions, i.e. when the size of the sample of extreme data for analysis is relatively small.

[Add New Section: P17 Line 319]
4.6 Model performance for smaller sample sizes
Here we repeat the analysis in Sections 4.1-4.5 above for the T = 100-year return value for SWH on the iso-depth contour and line transect, based on T0 = 50 years of data. The typical number of tropical cyclone events occurring

in 50 years is approximately 30, already corresponding to a very small sample size for extreme value analysis. We retain the largest n values of STM in the sample, for n = 10, 15 and 20 for this analysis. The overall performance of STM-E estimates, relative to those from single-location analysis and an empirical estimate from the full synthetic cyclone data is summarised in Figure 12 for the line transect, using the method of probability weighted moments (and see also Table 2 in the next section for a summary including estimates using maximum likelihood). The figure's features are similar to those of figures discussed earlier. Estimates from STM-E show lower bias and reduced uncertainty relative to those from single location analysis. There is slight underestimation of the return value, but the empirical estimate sits comfortably within the 25%-75% uncertainty band (corresponding to the "box" interval). The corresponding plots (not shown) for the iso-depth contour, and for estimation using maximum likelihood, are similar.

[Add new Table 2]

**Specific comments:**

[R2-1] P2 line 32 'Return level'. Most of the paper uses the term 'Return value' and, for clarity I think it would be better to be consistent throughout

>> *Thank you for the suggestion. We have decided to use the term "return value". We have modified the manuscript to be consistent throughout.*

[R2-2] P6 Figure 3 From the size of SWH it would appear that the SWAN model is using Wu (1982) wind drag formula which is linear with wind speed. Later versions of SWAN use Zijlema et al (2012) where the wind drag coefficient rises to a maximum and falls at high wind speeds. It would be worth specifying the version of SWAN used.

>> *Thank you for the comment. In our version of SWAN, the Wu (1982) wind drag formula is indeed selected, but with a prescribed maximum value of Cd=0.0035, so that the drag coefficient does not keep increasing for extreme events. This parameterization was found to give good results for the hurricanes we tested, although we must admit we could not find any in-situ data for validation in the most extreme conditions, so that the maximum HS for the strongest events found in the synthetic 3000-years equivalent database may indeed be overestimated. However, a proper representation of sea states in the most extreme cyclonic conditions remains a huge challenge for the scientific community as pointed out by a number of authors (e.g Bryant and Akbar, 2016), so that we can hardly select another formulation at present and claim that it would lead to better results. The second order polynomial fit proposed by Zijlema et al (2012) was tested for example for one category 4 hurricane, but the HS were clearly underestimated, so we decided to discard it for this study.*
*We have added the following comment in the manuscript.*

*Bryant, K.M., Akbar, M. (2016). An exploration of wind stress calculation techniques in hurricane storm surge modeling. Journal of Marine Science and Engineering , 4, 58. DOI: 10.3390/jmse4030058*

*Wu ,J. (1982). Wind-stress coefficients over sea surface from breeze to hurricane , Journal of Geophysical Research, 87, C12, 9704-9706.*

*Zijlema, M., G. Ph. Van Vledder and L.H. Holthuijsen (2012). Bottom friction and wind drag for spectral wave models,Coastal Engineering, 65,19-26.*

[Modified P4 Line 94 and add citation]
Here, the Wu (1982) wind drag formula is indeed selected, but with a prescribed maximum value of Cd=0.0035.

*Wu ,J. (1982). Wind-stress coefficients over sea surface from breeze to hurricane, Journal of Geophysical Research, 87, C12, 9704-9706.*

[R2-3] P7 line 142 I don't understand 'longitude-latitude transects with arbitrary orientation'. Please clarify.

>> *Thank you for the comment. We have modified the manuscript for clarification.*

[Modified P7 Line 148]
For example, the absence of a spatial trend in STM over the region can be assessed by quantifying the size of linear trends in STM along transects with arbitrary orientation in the region.

[R2-4] P9 lines 202-203 Please consider rephrasing this sentence to avoid a double negative. It is a bit confusing.

>> *Thank you for the comment and sorry for the confusion. We have revised the manuscript to clarify our message. STM-E only works when STM and exposure are exchangeable. The location of STM indicates where the peak of exposure is. For example, if there is an obvious spatial trend in STMs, the assumption of exchangeability is denied. Such exchangeability is assessed comprehensively using the rank correlation with Kendall's tau in this paper.*

[Modified P9 Line 208]
There is no obvious spatial dependence between the size of STM and its location

[R2-5] P11 line 229-231 If I understand correctly, you have modelled 1971 cyclones so would it not be better to count back rather than randomly sample? It would be a more robust way of estimating the 500 year return value.

>> *Thank you for the comment. We agree with your suggestion. We deleted the black boxplot and put a single black line in the figure indicating the empirical estimate of the 500-year return value that was derived from the full data set.*
*In addition, the performance of the estimation improved as we modified how we estimate the true value. Several modifications were made to the manuscript correspondingly.*

[Modified Figure. 6, 7, 8, 9 and 12 (new figure) and its captions]

[Modified Table 1]

[Modified Section 4.2 to explain procedure]
P12 Line 242: Thus, we are able to estimate the 500-year return value at any location using the full synthetic cyclone data, by simply interpolating the 6th and 7th largest values, corresponding to the non-exceedance probability in 500 years.

[Modified Section 4.3 & 4.4 to update results]
P12 Line 261: Bias characteristics for single-location and STM-E estimates are relatively similar in general. It can be seen however (from the longer red whiskers) that the uncertainty in single-location estimates is greater in general from the corresponding STM-E estimates.

P13 Line 280: There is some evidence that the STM-E median estimate increases with increasing sample size, and that this reduces bias.

[R2-6] P14-15 Figures 6 to 9 There is a lot of information included in the graphs and they are hard to interpret. It is difficult to compare black with blue and red. They need to be bigger which may require splitting into smaller subsets.

*>> Thank you for the comment. We agree that the figure had too much information and was difficult to read. We have made the following modifications for clarification.*
- *We deleted the black boxplot and put a single black line in the figure indicating the empirical estimate of the 500-year return value that was derived from the full data set (regarding your previous comment).*
- *We removed the green line indicating the depth as it could be explained in the caption of Figure 2.*

[Modified caption of Figure 2]
Water depths along the line transect with locations numbered 20-31 are 58m, 235m, 543m, 754m, 819m, 1206m, 1513m, 2350m, 3265m, 4059m, 5074m, 5586m.

[Modified Figure. 6, 7, 8, 9 and 12 (new figure) and its captions]

[R2-7] P14 line 271 Should this say decreasing rather than increasing? Either way perhaps this sentence needs some clarification

*>> Thank you for the comment. It is "increasing" but we agree it is difficult to understand. We have made the following changes for clarification.*

[Modified P13 Line 283]
Comparing box-whisker plots from centre to left for each location in the figures in Sections 4.3 and 4.4 suggests that there is sometimes a small increasing trend in return value estimates from STM-E as a function of increasing sample size for inference.

[R2-8] P15 figure 11 Red and blue show a consistent difference in shape for all sample sizes. Is this important and is it common to other locations?

*>> Thank you for the comment. The consistent behaviour of STM-E seems to come from STM, which is an interesting result. In Figure 10, we have summarised the slight under-estimation of STM for ML. As for PWM, the estimation is not biased but shows a positive trend as sample size increase. This is also seen in the STM-E results. In addition, the performance of the estimation improved as we modified how we estimate the true value. Several modifications were made to the manuscript correspondingly.*
*Following reviewer #1's comment, we have thoroughly investigated the effect of exposure is independent of STM magnitude. Thus, we assume the STM-E is mainly influence by the estimation of STMs.*
*We have added the following comment in the manuscript.*

[Modified P15 Line 285]

We investigate the trend further here. Figure 10 gives estimates for the 500-year return value of space-time maximum STM (as opposed to the full STM-E estimate for SWH) as a function of sample size used for estimation, using maximum likelihood estimation (blue) and probability weighted moments (red). Also shown in black is the empirical estimate of the 500-year STM return value obtained directly from the synthetic cyclone data. The figure shows a number of interesting effects. Firstly, STM estimates from both maximum likelihood and the probability weighted moments increase with increasing sample size n, and that this effect is more pronounced for probability weighted moments. As a result, the bias of estimates using probability weighted moments is considerably larger than that from maximum likelihood estimation for sample size of 60. The uncertainty of estimates from probability weighted moments is also somewhat larger than that from STM-E.

[Modified P16 Line 317]
Since exposure distribution at a location is not dependent on the magnitude of STM, we assume that the overall performance of STM-E is mainly governed by the estimation of STM.

[R2-9] P18 line 313 Is this the case for both MLE and PWM? It doesn't look that way.

>> *Thank you for the comment. We have reviewed the results and modified the manuscript for clarification. The number of sample size from center to outside. For STM-E, it is center to left.*

[Modified P13 Line 283]
Comparing box-whisker plots from centre to left for each location in the figures in Sections 4.3 and 4.4 suggests that there is sometimes a small increasing trend in return value estimates from STM-E as a function of increasing sample size for inference.

**Typos:**
[R2-1] P6 line 116 remove 'on'
>> *Thank you for the suggestion. We have modified the manuscript accordingly.*

[R2-1] P7 line 145 Insert 'be' between 'should' and 'sufficiently'
>> *Thank you for the suggestion. We have modified the manuscript accordingly.*

[R2-1] P9 line 199 An extra 'no' has been included. Please use the most appropriate of 'no obvious' or 'obviously no'
>> *Thank you for the suggestion. We have modified the manuscript as follows.*

[Modified P9 Line 208]
There is no obvious spatial dependence between the size of STM and its location

[R2-1] P18 line 331 'that' should be 'than'
>> *Thank you for the suggestion. We have modified the manuscript accordingly.*

*We again thank the reviewer for his/her kind suggestion and insightful comments on the paper. We hope we have addressed your concerns adequately.*

[revised manuscript text omitted]

---

## Author Comment (AC2)

**Rebuttal for Reviewer #1**

**Dear Reviewer #1,**

**General comments**

The proposed manuscript deals with the estimation of return levels (or high quantiles) in regions affected by tropical cyclones. The paper focuses on using a synthetic database of cyclones to validate the proposed method, developed in other papers, namely the STM-E approach. The paper is well organized with clear motivations, sensible application of the method, and an important issue in tropical areas.

*>> Thank you for the encouraging comments. We have considered your suggestions for revision individually. Please find our correspondence below (in italic).* Red text *indicates the actual modification on the manuscript. In addition, the locations of modification are in blue text in the manuscript. We hope you find them acceptable.*

**Specific issues**

[R1-1] Near L.200, the sentence " There is no obviously no spatial dependence between the size of STM and its location" is quite unclear to me. The figure clearly shows that the locations of STM are unevenly distributed in space, in particular, the points are located on the boundary of the domain, although there is no discussion about this repartition.

*>> Thank you for the comment and sorry for the confusion. We have revised the manuscript to clarify our message. STM-E only works when STM and exposure are exchangeable. The location of STM indicates where the peak of exposure is. For example, if there is an obvious spatial trend in STMs, the assumption of exchangeability is violated. Such exchangeability is assessed using the rank correlation of STMs and exposures with Kendall's tau in this paper.*
*Thanks for the comment regarding STMs located on the boundary of the domain. As suggested, the true STM of some extreme event occurred outside the region. In this model, STM and exposure are defined within the limited region. Essentially, we are defining a conditional STM and conditional exposure given within this region. For cases where the STM is truncated, the exposure will have larger values. We have modified the manuscript to elaborate on this point.*

[Modified P9 Line 208]
There is no obvious spatial dependence between the size of STM and its location

[Modified P10 Line 215]
In this sense, we are performing the STM-E analysis conditional on the choice of region. For example, consider a cyclone for which the location of the STM value $s*$ falls outside the region of interest. Then the conditional STM value $s$ for the cyclone (within the region) will of course be smaller than $s*$; however, the cyclone's conditional exposure (assessed relative to the conditional STM $s$ for the region, rather than the full STM $s*$) will consequently be larger.

[R1-2] Figure 6, 7,8,9: Quite difficult to read... What do the black boxplots correspond to? I think if the whole database is considered, the 500-year return level should be only one value per location, with associated CI?

*>> Thank you for the comment. We agree that the figure had too much information and was difficult to read. We have made the following modification for clarification.*

- *We deleted the black boxplot and put a single black line in the figure indicating the empirical estimate of the 500-year return value that was derived from the full data set. This modification was based on a suggestion from Reviewer #2. The manuscript is revised accordingly with the updated true value, in general the performance of STM-E looks better.*
- *We removed the green line indicating the depth as it could be explained in the caption of Figure 2.*

[Modified caption of Figure 2]
Water depths along the line transect with locations numbered 20-31 are 58m, 235m, 543m, 754m, 819m, 1206m, 1513m, 2350m, 3265m, 4059m, 5074m, 5586m.

[Modified Figure. 6, 7, 8, 9 and 12 (new figure) and its captions]
[Modified Table 1]
[Modified Section 4.3 & 4.4 to update results]
P12 Line 261: Bias characteristics for single-location and STM-E estimates are relatively similar in general. It can be seen however (from the longer red whiskers) that the uncertainty in single-location estimates is greater in general from the corresponding STM-E estimates.
P13 Line 280: There is some evidence that the STM-E median estimate increases with increasing sample size, and that this reduces bias.

[R1-3] There is no clear dependence of the depth here, nor is it included in the model, so maybe the authors should comment on why they include this covariate in the figures, or if it is a perspective for future work.

*>> Thank you for the comment. In general, the behavior of wave height can be dependent on depth. Ocean waves in shallow water can be affected by ocean bottom, and the wind or wave propagation can be weakened in the vicinity of coastlines. Thus, we decided to validate the method for various depth conditions. The results show a positive correlation between depth and wave height. We have added the following comment on our motivation in the manuscript.*

[Modified P10 Line 220]
It is known that SWH at a location is dependent on water depth, bathymetry and coastlines, since e.g. ocean waves in shallow water are influenced by bottom effects, and since both wind and wave propagation can be weakened in the vicinity of coastlines.

[R1-1] In the paragraph starting L290, the authors claim that they study the relationship between STM and Exposure in Figure 11, but the figure only shows two distributions for maximal and minimal STM, maybe a more quantitative assessment with parametric models would help understanding the (absence of) relation.

*>> Thank you for the comment. We have considered how to assess the exposure as suggested. Our first choice would be to use rank correlation, but there is no straightforward technique to rank the exposure as it is a cumulative distribution function. Instead, we decided to show that the exposure cdfs corresponding to the large STM and small STM are not "not special" relative to all the other pairs of exposure cdfs using the Kullback–Leibler divergence to measure the difference between two exposure cdfs. We calculate KL for 1,000 sets of two*

*randomly-selected exposure cdfs and build a null distribution of exposure cdf pair variability. In addition, we calculate $KL^*$ between the exposure cdfs for the largest and smallest STM. If $KL^*$ is randomly distributed within the null distribution, this indicates there are no special characteristics in the exposure distribution with large or small magnitude of STM (Figure 11). Also, the quantile values of $KL^*$ in the corresponding null distribution for all locations and sample size are summarized. We can see they are uniformly distributed (validated with Kolmogorov-Smirnov test with p-value of around 0.6), suggesting the independence of STM and exposure cdfs. The results are summarized in a new Figure 11 with an example of null distribution at Location 21 together with the cdf showing uniform distribution.*

[Modified P16 Line 301]

One of the assumptions underpinning the STM-E approach is that the exposure distribution at a location is not dependent on the magnitude of STM. We investigate this further here. Our aim is to show that the empirical cumulative distribution function for exposure (henceforth ECDF for brevity) corresponding to the largest and smallest STMs are typical of ECDFs in general, and are in no way special relative to EDCFs corresponding to other cyclones. We can quantify the difference between two ECDFs using the Kullback–Leibler divergence (KL). We proceed to estimate the "null" distribution of KL using 1,000 sets of randomly-selected pairs of ECDFs. In addition, we calculate the Kullback-Leibler divergence (Liese and Vajda (2006)) $KL^*$ for the pair of ECDFs corresponding to cyclones with the largest and smallest STMs. If there is no dependence of ECDF on STM, then the value of $KL^*$ should correspond to a random draw from the null distribution of KL. The left-hand panel of Figure 11 illustrates the null distribution of KL at Location 21, for sample size 20, together with the corresponding value of $KL^*$. We note that the value of $KL^*$ is not extreme in the null distribution. In the right-hand panel of Figure 11, the empirical cumulative distribution function of the non-exceedance probability of $KL^*$ (in the corresponding null distribution) is estimated over all locations and sample sizes. The approximate uniform density found, suggests indeed that $KL^*$ corresponds to a random draw from the null distribution; a Kolmogorov-Smirnoff test on the data suggested that it was not significantly different to a random sample from a uniform distribution on [0,1]

Complementary analyses (not shown) evaluated $KL^*$ for ECDFs corresponding to the largest two STMs in the data, and (separately) for ECDFs corresponding to the smallest two STMs. Results again indicated that both of these choices for $KL^*$ could be viewed as random in their null distributions. Since exposure distribution at a location is not dependent on the magnitude of STM, we assume the overall performance of STM-E is mainly governed by the estimation of STM.

[Modified Figure 11 with caption]

Left: Histogram of KL from random pairs of empirical distribution functions for exposure (corresponding to the "null" distribution of KL), together with $KL^*$ for Location 21 with sample size 20. Right: Histogram of the non-exceedance probability of $KL^*$ (in the corresponding null distribution for KL) over all locations and sample sizes.

[R1-4] The authors look at a very high quantile (500-year return level), while it is of more common practice to estimate the 100-year return level: the conclusions may be rather less clear due to the uncertainties linked to estimation of such high quantiles.

*>> Thank you for the comment. We agree with your comment and decided to add an additional analysis of 100-year return level estimation. Our initial aim was to validate the methodology with long data, i.e. estimation with large enough samples. The case for 500-year return level estimation suggested that extracting the largest samples*

*from around ⅓ of the tropical cyclones during the observation period is necessary to eliminate non-extreme samples.*

*Based on this suggestion, we performance assessment of STM-E in a more realistic / practical situation, i.e. estimation of 100-year return level from 50 years data period choosing 10~ 20 top cyclones. The results for this analysis was generally the same as what we found in the 500yr return level case.*

*The results and discussion made from the analysis is provided in the manuscript as follows.*

[Modified Abstract: P1 Line 11]
similar results were found for estimation of the 100-year return value from samples corresponding to approximately 50 years of data.

[Modified Introduction (Objective and Layout): P3 Line 69]
This case will assess the performance of STM-E when reasonable sample sizes of extreme values are available for inference. In addition, we conduct the corresponding estimation for the $T = 100$- year return value for SWH, and its uncertainty, based on random samples of tropical cyclones corresponding to $T0 = 50$ years of observation. This case is to assess the performance of STM-E under practical conditions, i.e. when the size of the sample of extreme values for analysis is relatively small.

[Add New Section: P17 Line 319]
4.6 Model performance for smaller sample sizes
Here we repeat the analysis in Sections 4.1-4.5 above for the $T = 100$-year return value for SWH on the iso-depth contour and line transect, based on $T0 = 50$ years of data. The typical number of tropical cyclone events occurring in 50 years is approximately 30, already corresponding to a very small sample size for extreme value analysis. We retain the largest n values of STM in the sample, for $n = 10$, 15 and 20 for this analysis. The overall performance of STM-E estimates, relative to those from single-location analysis and an empirical estimate from the full synthetic cyclone data is summarised in Figure 12 for the line transect, using the method of probability weighted moments (and see also Table 2 in the next section for a summary including estimates using maximum likelihood). The figure's features are similar to those of figures discussed earlier. Estimates from STM-E show lower bias and reduced uncertainty relative to those from single location analysis. There is slight underestimation of the return value, but the empirical estimate sits comfortably within the 25%-75% uncertainty band (corresponding to the "box" interval). The corresponding plots (not shown) for the iso-depth contour, and for estimation using maximum likelihood, are similar.

[Add new Table 2]

[R1-5] There is constant under-estimation of the return levels as seen in the figures and table 1. Is there a way to decide if it does come from the STM part of the model or the Expose part? Maybe the authors could provide keys to understanding where do the limitations come from.

*>> Thank you for the comment. In Figure 10, we have summarised the slight under-estimation of STM for ML. As for PWM, the estimation is not biased but shows a positive trend as sample size increase. This is also seen in the STM-E results. In addition, the performance of the estimation improved as we modified how we estimate the true value on the suggestion of Reviewer2. Several modifications were made to the manuscript correspondingly.*

*Regarding your first comment, we have thoroughly investigated the effect of exposure is independent of STM magnitude. Thus, we assume the STM-E is mainly influence by the estimation of STMs.*
*We have added the following comment in the manuscript.*

[Modified P15 Line 285]
We investigate the trend further here. Figure 10 gives estimates for the 500-year return value of space-time maximum STM (as opposed to the full STM-E estimate for SWH) as a function of sample size used for estimation, using maximum likelihood estimation (blue) and probability weighted moments (red). Also shown in black is the empirical estimate of the 500-year STM return value obtained directly from the synthetic cyclone data. The figure shows a number of interesting effects. Firstly, STM estimates from both maximum likelihood and the probability weighted moments increase with increasing sample size n, and that this effect is more pronounced for probability weighted moments. As a result, the bias of estimates using probability weighted moments is considerably larger than that from maximum likelihood estimation for sample size of 60. The uncertainty of estimates from probability weighted moments is also somewhat larger than that from STM-E.

[Modified P16 Line 317]
Since exposure distribution at a location is not dependent on the magnitude of STM, we assume that the overall performance of STM-E is mainly governed by the estimation of STM.

**Technical corrections**

[R1-6] Figure 10: the 0 on the x-axis should be a 1,200? Again, how do you obtain the box-plot here?

*>> Thank you for the comment and sorry for the confusion. Following the modification made above, the true value is no longer a boxplot. The N-year return period value is now derived from the synthetic data set as a point value and now depicted as a black dashed line.*

[Modified Figure 10 and caption]

[R1-7] Table 1 may be more clear if the number of observations is replaced by the corresponding years.

*>> Thank you for the comment and sorry for the confusion. The results depicted in Table 1 all correspond to 200-year data set. The number of samples (i.e. n = 20, 30, 40, 50, 60) indicates how many samples we extract from the data set, corresponding to lowering the extreme value threshold. The same is done for analysis of 100-year return period value. We have clarified the manuscript to make our objectives clear.*

[Modified P11 Line 232]
(corresponding to lowering the extreme value threshold)

[Modified caption of Table 1]

*We again thank the reviewer for his/her kind suggestion and insightful comments on the paper. We hope we have addressed your concerns adequately.*

[revised manuscript text omitted]

---

## Author Response (AR2)

**Rebuttal for Reviewer**

**Dear Reviewer,**

Suggestions for revision or reasons for rejection (will be published if the paper is accepted for final publication) Minor errors:

*>> Thank you for the thorough review. We have considered your comments for revision individually. Please find our correspondence below (in italic).* Red text *indicates the actual modification on the manuscript. In addition, the locations of modification are in blue text in the manuscript. We hope you find them acceptable.*

Figures 6 & 8: I think the y-axis should be labelled SWH (m)
*>> Thank you for the suggestion. We have revised the corresponding figures.*

Figure 11: the righthand graph is not a histogram.
*>> Thank you for the suggestion. We have revised the caption of the corresponding figure..*

[Modified Caption of Figure 11]
Figure 11. Left: Histogram of KL from random pairs of empirical distribution functions for exposure (corresponding to the "null" distribution of KL), together with KL∗ for Location 21 with sample size 20. Right: QQ-plot of the non-exceedance probability of KL∗ (in the corresponding null distribution for KL) over all locations and sample sizes.

Small addition:
Page 16: Please add a sentence noting that all locations are in water deep enough that waves are not be breaking. In shallow water, where waves are subject to breaking, it would be expected that exposure distribution would be dependent on STM and therefore this method would not be valid.
*>> Thank you for the suggestion. We agree with your comment and added the following explanation on the validity of the method.*

[Modified P16 Last Line]

[revised manuscript text omitted]